# Diffeomorphic Temporal Alignment Nets

**Ron Shapira Weber**
Ben-Gurion University
ronsha@post.bgu.ac.il

**Matan Eyal**
Ben-Gurion University
mataney@post.bgu.ac.il

**Nicki Skafte Detlefsen**
Technical University of Denmark
nsde@dtu.dk

**Oren Shriki**
Ben-Gurion University
shrikio@bgu.ac.il

**Oren Freifeld**
Ben-Gurion University
orenfr@cs.bgu.ac.il

## Abstract

Time-series analysis is confounded by nonlinear time warping of the data. Traditional methods for joint alignment do not generalize: after aligning a given signal ensemble, they lack a mechanism, that does not require solving a new optimization problem, to align previously-unseen signals. In the multi-class case, they must also first classify the test data before aligning it. Here we propose the Diffeomorphic Temporal Alignment Net (DTAN), a learning-based method for time-series joint alignment. Via flexible temporal transformer layers, DTAN learns and applies an input-dependent nonlinear time warping to its input signal. Once learned, DTAN easily aligns previously-unseen signals by its inexpensive forward pass. In a single-class case, the method is unsupervised: the ground-truth alignments are unknown. In the multi-class case, it is semi-supervised in the sense that class labels (but not the ground-truth alignments) are used during learning; in test time, however, the class labels are unknown. As we show, DTAN not only outperforms existing joint-alignment methods in aligning training data but also generalizes well to test data. Our code is available at https://github.com/BGU-CS-VIL/dtan.

## 1 Introduction

Time-series data often presents a significant amount of misalignment, also known as nonlinear time warping. To fix ideas, consider ECG recordings from healthy patients during rest. Suppose that the signals were partitioned correctly such that each segment corresponds to a heartbeat and that these segments were resampled to have equal length (*e.g.*, see Figure 1). Each resampled segment is then viewed as a distinct signal. The sample mean of these usually-misaligned signals (even when restricting to single-patient recordings) would not look like the iconic ECG sinus rhythm; rather, it would smear the correct peaks and valleys and/or contain superfluous ones. This is unfortunate as the sample mean, a cornerstone of Descriptive Statistics, has numerous applications in data analysis (*e.g.*, providing a succinct data summary). Moreover, even if one succeeds somehow in aligning a currently-available recording batch, upon the arrival of new data batches, the latter will also need to be aligned; *i.e.*, one would like to generalize the inferred alignment from the original batch to the new data without having to solve a new optimization problem. This is especially the case if the new dataset is much larger than the original one; *e.g.*, imagine a hospital solving the problem once, and then generalizing its solution, essentially at no cost, to align all the data collected in the following year. Finally, these issues become even more critical for multi-class data (*e.g.*, healthy/sick patients), where only in the original batch we know which signal belongs to which class; *i.e.*, seemingly, the new data will have to be explicitly classified before its within-class alignment.

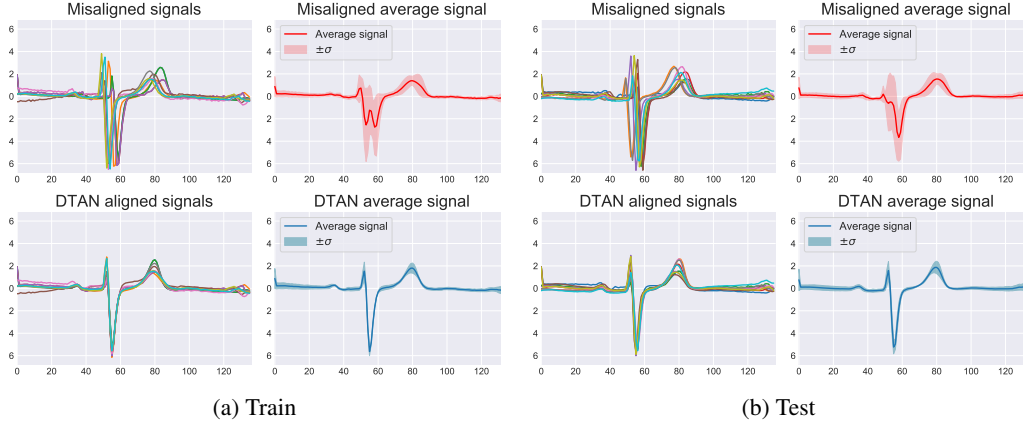

(a) Train                             (b) Test

Figure 1: Learning to generalize time-series joint alignment from train to test signals on the ECGFive-Days dataset [8]. Top row: 10 random misaligned signals from each set and their respective average signal (shaded areas correspond to standard deviations). Bottom: The signals after the estimated alignment. DTAN aligns, in an input-dependent manner, a new test signal in a single forward pass.

Let $(\boldsymbol{U}_i)_{i=1}^N$ be a set of $N$ time-series observations. The nonlinear misalignment can be written as:

$$(\boldsymbol{U}_i)_{i=1}^N = (\boldsymbol{V}_i \circ W_i)_{i=1}^N \tag{1}$$

where $\boldsymbol{U}_i$ is the $i^{\text{th}}$ misaligned signal, $\boldsymbol{V}_i$ is the $i^{\text{th}}$ latent aligned signal, "$\circ$" stands for function composition, and $W_i$ is a latent warp of the domain of $\boldsymbol{V}_i$. For technical reasons, the misalignment is usually viewed in terms of $T_i \triangleq W_i^{-1}$, the inverse warp of $W_i$, implicitly suggesting $W_i$ is invertible. It is typically assumed that $(T_i)_{i=1}^N$ belong to $\mathcal{T}$, some nominal family of warps parameterized by $\boldsymbol{\theta}$:

$$(\boldsymbol{V}_i)_{i=1}^N = (\boldsymbol{U}_i \circ T^{\boldsymbol{\theta}_i})_{i=1}^N, \quad T_i = T^{\boldsymbol{\theta}_i} \in \mathcal{T} \; \forall i \in (1,\dots,N). \tag{2}$$

The nuisance warps, $(T^{\boldsymbol{\theta}_i})_{i=1}^N$, create a fictitious variability in the range of the signals, confounding their statistical analysis. Thus, the *joint-alignment* problem, defined below, together with the ability to use its solution for generalization, is of great interest to the machine-learning community as well as to other fields.

**Definition 1 (the joint-alignment problem)** *Given* $(\boldsymbol{U}_i)_{i=1}^N$, *infer the latent* $(T^{\boldsymbol{\theta}_i})_{i=1}^N \subset \mathcal{T}$.

*We argue that this problem should be seen as a learning one, mostly due to the need for generalization.* Particularly, we propose a novel deep-learning (DL) approach for the joint alignment of time-series data. More specifically, inspired by computer-vision and/or pattern-theoretic solutions for misaligned images (*e.g.*, congealing [38, 31, 26, 25, 10, 11], efficient diffeomorphisms [19, 20, 56, 57], and spatial transformer nets [28, 32, 49]), we introduce the Diffeomorphic Temporal Alignment Net (DTAN) which learns and applies an input-dependent diffeomorphic time warping to its input signal to minimize a joint-alignment loss and a regularization term. In the single-class case, this yields an unsupervised method for joint-alignment learning. For multi-class problems, we propose a semi-supervised method which results in a single net (for all classes) that learns how to perform, within each class, joint alignment without knowing, at test time, the class labels. We demonstrate the utility of the proposed framework on both synthetic and real datasets with applications to time-series joint alignment, averaging and classification, and compare it with DTW Barycenter Averaging (DBA) [44] and SoftDTW [12]. On training data, DTAN outperforms both. More importantly, it generalizes to test data (and in fact excels in it); this is an ability not possessed by those methods.

**Our key contributions are as follows.** 1) DTAN, a new DL framework for learning joint alignment of time-series data; 2) A recurrent version of DTAN (which is also the first recurrent diffeomorphic transformer net); 3) A new and fast tool for averaging misaligned single-class time-series data; 4) The proposed learning-based method generalizes to previously-unseen data; *i.e.*, unlike existing methods for time-series joint alignment, DTAN can align new test signals and the test-time computations are remarkably fast.

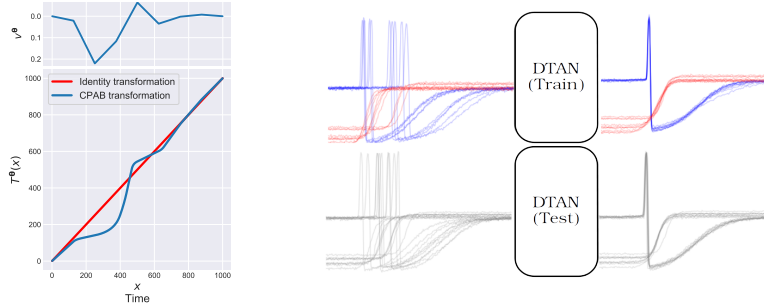

Figure 2: Left: An illustration of a CPAB warp (relative to the identity transformation) with its corresponding CPA velocity field (above). Right: DTAN joint alignment demonstrated on two classes of the Trace dataset [8]. During test, the class labels are unknown.

## 2   Related Work

**Dynamic Time Warping (DTW).** A popular approach for aligning a time-series pair is DTW [47, 48] which, by solving Bellman's recursion via dynamic programming, finds an optimal monotonic alignment between two signals. DTW does not scale well to the joint-alignment problem: computing a pairwise DTW for $N$ signals of length $K$ requires $O(K^N)$ operations [52], which is intractable for either a large $N$ or a large $K$. Moreover, averaging under the DTW distance is a nontrivial task, as it involves solving the joint-alignment problem. While several authors proposed smart solutions for the averaging problem [50, 22, 44, 43, 13, 12], none of them offered a generalization mechanism – that does not require solving a new optimization problem each time – for aligning new signals.

**Congealing, Joint Alignment, and Atlas-based Methods.** A congealing algorithm solves iteratively for the joint alignment (of a set of signals such as images, time series, *etc.*) by gradually aligning one signal towards the rest [31]. Typical alignment criteria used in congealing are entropy minimization [38, 31, 26, 37] or least squares [10, 11]. Also related is the Continuous Profile Model [33], a generative model in which each observed time series is a non-uniformly subsampled version of a single latent trace. While not directly related to our work, note that many medical-imaging works focus on building an atlas, including with diffeomorphisms (*e.g.*, [29]), via the (pairwise- or joint-) alignment of multiple images. Since all these methods above do not generalize, in order to align $N_{\text{test}}$ new signals to the average signal of the previously-aligned $N_{\text{train}}$ signals (or to an atlas), one must solve $N_{\text{test}}$ pairwise-alignment problems. Alternatively, to jointly align $N_{\text{test}}$ new signals, one must solve a new joint-alignment problem. In both cases, such solutions scale poorly with $N_{\text{test}}$. In the multi-class case, it is even worse since the new signals must be classified, and classification errors increase alignment errors. Note that in [25] the authors propose a two-step process: the first learns deep Convolutional Neural Networks (CNN) features, unrelated to alignment, and the second uses congealing to align these features (without learning how to align the features of a new data). In parallel to our work, and independently of it, Dalca *et al*. [14] propose a learning-based method for building deformable conditional templates based on diffeomorphisms. While their model offers generalization, they focus on neuroimaging and not time-series joint alignment.

**Spatial/Temporal Transformer Nets and Diffeomorphisms in DL.** In computer vision, the Spatial Transformer Net (STN) [28] was introduced to allow for invariances to spatial warps. While there are works on the pairwise alignment of time-series hidden states [50, 6, 21, 2], Temporal Transformer Nets (TTN), the time-series analog of STNs, were so far limited to affine transformations [41], phase and frequency offset recovery [42]. It was also proposed to use TTN on the 2D spectrogram of time series [58]. Very recently, Lohit *et al*. proposed a TTN based on 1D diffeomorphisms for time-series classification [35]; as their warps are not parametric, the method does not scale well with the signal's length; *e.g.*, a one-second input signal at 8kHz will yield a TTN with a final fully-connected (FC) layer of $dim = 8,000$ neurons, which in turn produces $8,000$ trainable weights per neuron in the previous layer (for comparison, we use an FC layer of $dim = 32$); moreover, the nonparametric form prevents them from having an equivalent to the efficient gradient that we use. In addition, none of these methods utilized TTN for learning time-series joint alignment.

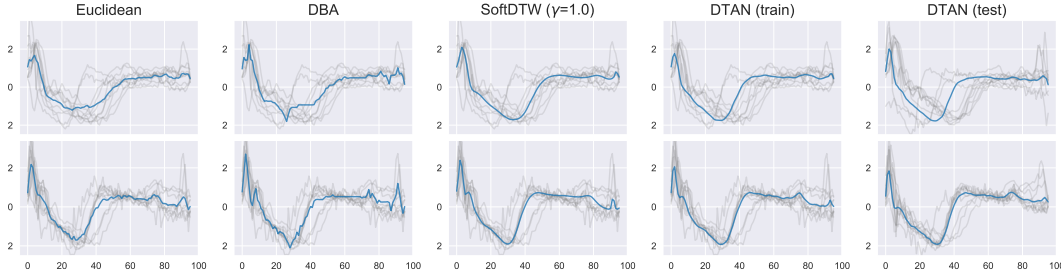

Figure 3: Time-series averaging methods comparison on the ECG200 dataset (each row depicts a different class). The Euclidean mean serves as a baseline, showing how nonlinear misalignment of the data confounds its averaging. Comparing with DTW-based methods, DTAN outperforms DBA on both train/test data. While the barycenter obtained by SoftDTW ($\gamma = 1$) is comparable to the one obtained by DTAN, it is (1) inapplicable to new signals; (2) computed on each class individually. DTAN, however, was trained on both classes together and generalized to test data (rightmost panels).

Recently, Skafte *et al.* [49] showed it is possible to explicitly incorporate flexible and efficient diffeomorphisms [19, 20] within DL architectures via an STN; particularly, they focused on image recognition and classification and their framework was supervised. Inspired by [49], we propose to use a diffeomorphic TTN to solve the joint-alignment problem. Our approach differs from [49] in the following: the signal type (1D signals vs. 2D images); the task (joint alignment vs. classification); amount of supervision (unsupervised/semi-supervised vs. supervised); usage of recurrent nets and warp regularization (here we use both, neither was used in [49]). In addition to [49], there are several works, particularly in medical imaging, that involve DL and diffeomorphisms. Their formulation is different from ours. E.g, while Yang *et al.* [55] use supervised DL to *predict* diffeomorphisms, their net has no STN so the diffeomorphisms are not explicitly incorporated in it. In contrast, unsupervised diffeomorphic alignment was achieved via an STN [15, 7]. In all these three works [55, 15, 7] (as well as in others omitted here due to space limits) the nets learn pairwise alignments, not joint alignment. In any case, we are unaware of works that use diffemorphic nonlinear transformer nets for *time-series* data (with the exception of [35]), let alone for joint alignment of such data (with no exceptions).

## 3   Preliminaries: Temporal Transformer Nets and Diffeomorphisms

**Temporal Transformer Nets.** Given $\mathcal{T}$, a spatial-warp family parameterized by $\boldsymbol{\theta}$, a Spatial Transformer (ST) layer performs a learnable input-dependent warp [28]. Reducing this from images (a 2D domain) to time series (1D), one obtains a TT layer (a TTN is a neural net with at least one TT layer). In more detail, let $\boldsymbol{U}$ denote the input of the TT layer. Its output consists of $\boldsymbol{\theta} = f_{\text{loc}}(\boldsymbol{U})$ and $\boldsymbol{V} = \boldsymbol{U} \circ T^{\boldsymbol{\theta}}$ (the latter, *i.e.*, the warped signal, is what is being passed downstream the TTN), where $T^{\boldsymbol{\theta}} \in \mathcal{T}$ is a 1D warp parameterized by $\boldsymbol{\theta}$. The function $f_{\text{loc}} : \boldsymbol{U} \mapsto \boldsymbol{\theta}$ is itself a neural net called the localization net. Let $\boldsymbol{w}$ denote the parameters (also known as weights) of $f_{\text{loc}}$ and let

$$F((\boldsymbol{U}_i, \boldsymbol{\theta}_i(\boldsymbol{U}_i; \boldsymbol{w}))_{i=1}^N) \tag{3}$$

denote a loss function. The TT layer is trained (*i.e.*, optimized over $\boldsymbol{w}$) along with the rest of the TTN. As is usual in DL, this involves back-propagation [46] which requires certain partial derivatives (see our **Sup. Mat.**). Also note one of these derivatives, $\nabla_{\boldsymbol{\theta}}(T^{\boldsymbol{\theta}}(\cdot))$, depends on the choice of $\mathcal{T}$.

**Diffeomorphisms.** As mentioned in § 1, $\mathcal{T}$ needs to be specified. In the context of time warping, *diffeomorphisms* is a natural choice [39]. A ($C^1$) diffeomorphism is a differentiable invertible map with a differentiable inverse. Working with diffeomorphisms usually involves expensive computations. In our case, since the proposed method explicitly incorporates them in a DL architecture, it is even more important (than in traditional non-DL applications of diffeomorphisms) to drastically reduce the computational difficulties: in training, evaluations of $x \mapsto T^{\boldsymbol{\theta}}(x)$ and $x \mapsto \nabla_{\boldsymbol{\theta}} T^{\boldsymbol{\theta}}(x)$ are computed at multiple time points $x$ and for multiple $\boldsymbol{\theta}$'s. Thus, until recently, explicit incorporation of highly-expressive diffeomorphism families into DL architectures used to be infeasible. This, however, is starting to change (*e.g.*, [49, 7]). Particularly, Skafte *et al.* [49] utilized, in their STNs, the CPAB warps that had been proposed by Freifeld *et al.* [19, 20] and are also used in this work. CPAB warps

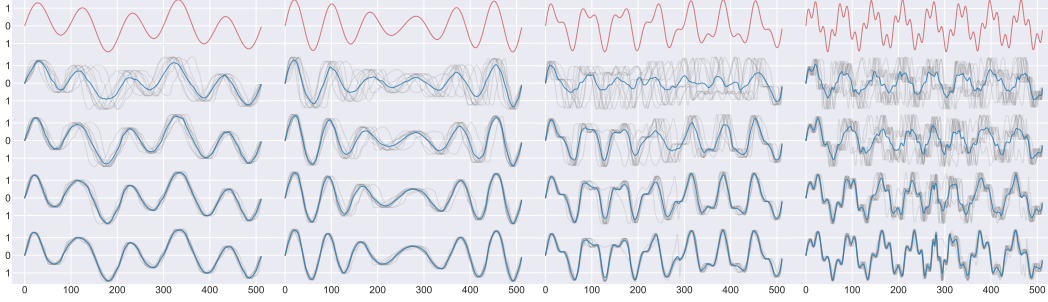

Figure 4: R-DTAN joint-alignment of synthetic data. Each column depicts a different class. Top row: Source latent signals from which each class was created. Second: 10 perturbed signals and their respective mean. Last three rows illustrate R-DTAN output at each recurrence, eventually unwarping the nonlinear misaligned applied to the latent source signals. All the results shown here are on test data, and were obtained by the same single net (without knowing, at test time, the class labels).

combine expressiveness and efficiency, making them a natural choice in a DL context [24, 49]. Other efficient and expressive diffeomorphisms (*e.g.*,[57, 4, 17, 3]) can also be explored in the DTAN context, provided they also offer an efficient and highly-accurate way to evaluate $x \mapsto \nabla_{\boldsymbol{\theta}} T^{\boldsymbol{\theta}}(x)$ as CPAB warps do [18]. Below we briefly explain CPAB warps (restricting the discussion to 1D, which is the domain of interest in this work), and refer the reader to [19, 20, 18] for more details. The name CPAB, short for CPA-Based, is due to the fact that these warps are based on Continuous Piecewise-Affine (CPA) velocity fields. The term "piecewise" is w.r.t. a partition, denoted by $\Omega$, of the signal's domain into subintervals. Let $\mathcal{V}$ denote the linear space of CPA velocity fields w.r.t. such a fixed $\Omega$, let $d = \dim(\mathcal{V})$, and let $v^{\boldsymbol{\theta}} : \Omega \to \mathbb{R}$, a velocity field parameterized by $\boldsymbol{\theta} \in \mathbb{R}^d$, denote the generic element of $\mathcal{V}$, where $\boldsymbol{\theta}$ stands for the coefficient w.r.t. some basis of $\mathcal{V}$. The corresponding space of CPAB warps, obtained via integration of elements of $\mathcal{V}$, is

$$\mathcal{T} \triangleq \{T^{\boldsymbol{\theta}} : x \mapsto \phi^{\boldsymbol{\theta}}(x; 1) \text{ s.t. } \phi^{\boldsymbol{\theta}}(x; t) = x + \int_0^t v^{\boldsymbol{\theta}}(\phi^{\boldsymbol{\theta}}(x; \tau)) \, \mathrm{d}\tau \text{ where } v^{\boldsymbol{\theta}} \in \mathcal{V}\}; \qquad (4)$$

it can be shown that these warps are indeed ($C^1$) diffeomorphisms [19, 20]. See Figure 2 for a typical warp. While $v^{\boldsymbol{\theta}}$ is CPA, $T^{\boldsymbol{\theta}} : \Omega \to \Omega$ is not (*e.g.*, $T^{\boldsymbol{\theta}}$ is differentiable). CPA velocity fields support an integration method that is faster *and* more accurate than typical velocity-field integration methods [19, 20]. The fineness of $\Omega$ controls the trade-off between expressiveness of $\mathcal{T}$ on the one hand and the associated computational complexity and dimensionality on the other hand. Importantly in the TTN context, the *CPAB gradient*, $\nabla_{\boldsymbol{\theta}} T^{\boldsymbol{\theta}}(x)$, is given by the efficient solution of a system of coupled integral equations [20]; see [18] for details.

## 4 The Proposed Diffeomorphic Temporal Alignment Nets

Definition 1 requires the specification of $\mathcal{T}$ and a loss function for estimating $(T^{\boldsymbol{\theta}_i})_{i=1}^N$. To meet our goal, *i.e.*, solving the joint-alignment problem while being able to generalize its solution to the alignment of new data, we propose a DL-based method which includes a TTN with diffeomorphic TT layers. Particularly, we choose $\mathcal{T}$ to be a family of 1D CPAB warps [19, 20] and incorporate the latter within TT layers. For simplicity, we base the data term of the training loss on least squares but other criteria can be used as well. Altogether, this lets us propose the first DTAN for time-series joint alignment (it is also the first diffeomorphic transformer net for joint alignment of any kind of data, not just time series). Below we explain the method in more detail, including how it is used for aligning and averaging either existing or new data. We also discuss the critical role of warp regularization as well as recurrent DTANs.

**Time-series Joint Alignment.** Let $\boldsymbol{U}_i$ denote an input signal, let $\boldsymbol{\theta}_i = f_{\mathrm{loc}}(\boldsymbol{U}_i, \boldsymbol{w})$ denote the corresponding output of the localization net $f_{\mathrm{loc}}(\cdot, \boldsymbol{w})$ of weights $\boldsymbol{w}$, and let $\boldsymbol{V}_i$ denote the result of warping $\boldsymbol{U}_i$ by $T^{\boldsymbol{\theta}_i} \in \mathcal{T}$; *i.e.*, $\boldsymbol{V}_i = \boldsymbol{U}_i \circ T^{\boldsymbol{\theta}_i}$, where $\boldsymbol{\theta}_i$ depends on $\boldsymbol{w}$ and $\boldsymbol{U}_i$, as defined above. Consider first the case where all the $\boldsymbol{U}_i$'s belong to the same class. As the variance of the observed $(\boldsymbol{U}_i)_{i=1}^N$ is (at least partially) explained by the latent warps, $(T^{\boldsymbol{\theta}_i})_{i=1}^N$, we seek to minimize the

empirical variance of the warped signals, $(V_i)_{i=1}^N$. In other words, our data term in this setting is

$$F_{\text{data}}\left(\boldsymbol{w}, (\boldsymbol{U}_i)_{i=1}^N\right) \triangleq \frac{1}{N}\sum_{i=1}^N \left\|\boldsymbol{V}_i(\boldsymbol{U}_i; \boldsymbol{w}) - \frac{1}{N}\sum_{j=1}^N \boldsymbol{V}_j(\boldsymbol{U}_j; \boldsymbol{w})\right\|_{\ell_2}^2 \qquad (5)$$

where $\|\cdot\|_{\ell_2}$ is the $\ell_2$ norm. Note this setting is unsupervised. For multi-class problems, our data term is the sum of the within-class variances:

$$F_{\text{data}}\left(\boldsymbol{w}, (\boldsymbol{U}_i)_{i=1}^N\right) \triangleq \sum_{k=1}^K \frac{1}{N_k}\sum_{i:z_i=k} \left\|\boldsymbol{V}_i\left(\boldsymbol{U}_i; \boldsymbol{w}\right) - \frac{1}{N_k}\sum_{j:z_j=k} \boldsymbol{V}_j(\boldsymbol{U}_j; \boldsymbol{w})\right\|_{\ell_2}^2 \qquad (6)$$

where $K$ is the number of classes, $z_i$ takes values in $\{1, \ldots, K\}$ and is the class label associated with $\boldsymbol{U}_i$ (namely: $z_i = k$ if and only if $\boldsymbol{U}_i$ belongs to class $k$), and $N_k = |\{i : z_i = k\}|$ is the number of examples in class $k$. This is a semi-supervised setting in the following sense: the labels, $(z_i)_{i=1}^N$ are known during the learning (but not during the test) while the within-class alignment remains unsupervised as in the single-class case. Importantly, note that the same single network is responsible for aligning each of the classes; *i.e.*, $\boldsymbol{w}$ does not vary with $k$; see Figure 2. In both the single- and multi-class cases, we (unlike Skafte *et al.* [49]) also use a regularization term on the warps,

$$F_{\text{reg}}(\boldsymbol{w}, (\boldsymbol{U}_i)_{i=1}^N) = \sum_{i=1}^N (\boldsymbol{\theta}_i(\boldsymbol{w}, \boldsymbol{U}_i))^T \Sigma_{\text{CPA}}^{-1} \boldsymbol{\theta}_i(\boldsymbol{w}, \boldsymbol{U}_i) \qquad (7)$$

where $\Sigma_{\text{CPA}}$ is a CPA covariance matrix (proposed by Freifeld *et al.* [19, 20]) associated with a zero-mean Gaussian smoothness prior over CPA fields. Akin to the standard formulation in, *e.g.*, Gaussian processes [45], $\Sigma_{\text{CPA}}$ has two parameters: $\lambda_{\text{var}}$, which controls the overall variance, and $\lambda_{\text{smooth}}$, which controls the smoothness of the field. A small $\lambda_{\text{var}}$ favors small warps (*i.e.*, close to the identity) and vice versa; similarly, the larger $\lambda_{\text{smooth}}$ is, the more it favors CPA velocity fields that are almost purely affine and vice versa. This also gives another way, an alternative to changing the resolution of $\Omega$, to control the amount of expressiveness of the warps. In the context of our joint-alignment task (as opposed to, say, the classification task in [49]), using regularization is critical, partly since it is too easy to minimize $F_{\text{data}}$ by unrealistically-large deformations that would cause most of the inter-signal variability to concentrate on a small region of the domain; the regularization term prevents that. Our loss function, to be minimized over $\boldsymbol{w}$, is

$$F(\boldsymbol{w}, (\boldsymbol{U}_i)_{i=1}^N) = F_{\text{data}}(\boldsymbol{w}, (\boldsymbol{U}_i)_{i=1}^N) + F_{\text{reg}}(\boldsymbol{w}, (\boldsymbol{U}_i)_{i=1}^N). \qquad (8)$$

The optimization (*i.e.* the training of the net) is done via standard methods for DL training (see § 5).

**Generalization via the Learned Joint Alignment.** Once the net is trained, a signal $\boldsymbol{U}$ (regardless whether it is a training or a test signal) is aligned as follows. First set $\boldsymbol{\theta} = f_{\text{loc}}(\boldsymbol{U})$; *i.e.*, a forward pass of the net (an operation which is, as is usually the case in DL, simple and very fast). Next, obtain the aligned signal, $\boldsymbol{V}$, via warping $\boldsymbol{U}$ by $T^{\boldsymbol{\theta}}$; *i.e.*, set $\boldsymbol{V} = \boldsymbol{U} \circ T^{\boldsymbol{\theta}}$. Especially useful and elegant is the fact that, in the multi-class case, the same single net aligns each new test signal, without knowing the label of the latter. This is in sharp contrast to other joint-alignment methods (*e.g.*, those based on DBA, SoftDTW, atlases, *etc.*) that require knowing the label of the to-be-aligned signal.

**Time-series Averaging.** The data misalignment distorts, among other things, the sample mean [53, 23]. As discussed in § 2, averaging under the DTW distance is a common approach to this issue [44, 43, 13, 12]; however, such non-learning DTW-based methods are computationally expensive. This is especially problematic since, as these methods do not generalize, each batch of new signals requires them to solve another optimization problem. In contrast, since DTAN easily aligns new signals inexpensively and almost instantaneously via its forward pass, it also provides, in the single-class case, an instant mechanism for quickly averaging a new collection of previously-unseen signals (see Figure 3) by simply computing the sample mean of the warped test data: $\bar{\boldsymbol{V}} = \frac{1}{N}\sum_{j=1}^N \boldsymbol{V}_j(\boldsymbol{U}_j; \boldsymbol{w})$.

**Variable length and multi-channel data** The current work focuses on univariate time-series data and fixed-length input. The generalization to multichannel signal is trivial: DTAN can either apply the same warp to all channels (just like an STN warps RGB images) or learn and apply different warps for each channel. To generalize DTAN for variable length (VL) input, we need to consider $f_{\text{loc}}$, $\mathcal{T}$ and the loss function. For $f_{\text{loc}}$, Recurrent Neural Networks (RNNs) are a natural choice, as they are designed to handle VL inputs. A nominal CPAB family, $\mathcal{T}$, is capable of warping any time interval towards any other, even if they are of different lengths, as long as no boundary conditions are used. Finally, a loss function that can handle VL must be chosen (*e.g.*, SoftDTW [12]).

Table 1: Synthetic data variance of the misaligned data ("Baseline") and the aligned data via DTAN, Recurrent-DTAN (R-DTAN2 and 4). For each set, $\mathrm{Dir}(k)$, $k$ specifies the seriousness of the deformation, where a lower $k$ indicates higher deformations. DTAN exhibits comparable results in terms of variance reduction between the train and test sets. Increasing the number of applied warps via an R-DTAN (without increasing the number of learned parameters) further decreases the variance.

| | Train set variance | | | | Test set variance | | | |
|---|---|---|---|---|---|---|---|---|
| Dataset | Baseline | DTAN | R-DTAN2 | R-DTAN4 | Baseline | DTAN | R-DTAN2 | R-DTAN4 |
| $\mathrm{Dir}(32)$ | 0.483 | 0.136 | 0.106 | **0.088** | 0.466 | 0.234 | 0.167 | **0.130** |
| $\mathrm{Dir}(16)$ | 0.522 | 0.240 | 0.162 | **0.098** | 0.514 | 0.332 | 0.24 | **0.154** |
| $\mathrm{Dir}(8)$ | 0.536 | 0.254 | 0.181 | **0.122** | 0.532 | 0.362 | 0.248 | **0.183** |

**Recurrent DTANs.** While often a coarse $\Omega$ suffices, the expressiveness of $\mathcal{T}$ can be increased using a finer $\Omega$ at the cost of computation speed and a higher $d$ [19, 20]. In fact, at the limit of an infinitely-fine $\Omega$, any diffeomorphism that is representable by integrating a Lipshitz-continuous stationary velocity field can be approximated by a CPAB diffeomorphism [19, 20]. Moreover, CPAB warps do not form a group under the composition operation [20] (even though they contain the identity warp and are closed under inversion); *i.e.*, the composition of CPAB warps is a diffeomorphism but usually not CPAB itself. Thus, a way to increase expressiveness without refining $\Omega$ is by composing CPAB warps [20]. Concatenating CPAB warps increases expressiveness beyond $\mathcal{T}$ as it implies a non-stationary velocity field which is CPA w.r.t. $\Omega$ and piecewise constant w.r.t. time. Compositions increase dimensionality, but the overall cost of evaluating the composed warp scales better (in comparison with refinement of $\Omega$), and it is also easier to infer the $\boldsymbol{\theta}$'s. While this fact was not exploited in [49], we leverage it here as follows. We propose the Recurrent-DTAN (R-DTAN), a net that recurrently applies nonlinear time warps, via diffeomorphic TT layers, to the input signal (Figure 4). By sharing the learned parameters by all the TT layers, an R-DTAN increases expressiveness without increasing the number of parameters. While this is similar to, and inspired by, how Lin *et al*. [32] use a recurrent net with affine 2D warps, there is a key difference: since in the affine case zero-boundary conditions imply degeneracies, they explained they had to propagate warp parameters instead of the warped image as they would have liked. In contrast, as CPAB warps support optional zero-boundary conditions, propagating a warped signal through an R-DTAN is a non-issue.

**Implementation.** We adapted, to the 1D case, the implementation from [16] of the CPAB transformer layer, CPAB gradient, the Tensorflow C++ API, and Keras wrapper for the transformer layer. We also implemented in Tensorflow/Keras the CPAB regularization term as well as the recurrent net, both of which were not used in [49]. To summarize, users can benefit from our DTAN implementation in any Tensorflow [1] or Keras [9] generic DL architecture in a few lines of code.

## 5 Experiments and Results

We evaluated DTAN's time-series joint alignment of both synthetic and real-world data. For simplicity, in our experiments $f_{\mathrm{loc}}$ is set to be a 1D CNN consisting of 3 conv-layers (128–64–64 filters per layer, respectively) each followed by a ReLU nonlinear activation function [40], batch-normalization and max-pooling layers [27], where $d = \dim(\boldsymbol{\theta}) = 32$. The learning rate was $\eta = 10^{-4}$, set to minimize Eq. (6) via the Adam optimizer [30]. The last activation function was $tanh$.

### 5.1 Learning Joint Alignment of Synthetic Data

We generated synthetic data by perturbing 4 synthetic signals using random warps sampled from a Dirichlet prior (see **Sup. Mat.** for details of the data-generation procedure). We generated 250 samples per-class (1000 in total) and used a 60-20-20% train, validation and test split, choosing the model with the lowest validation loss (where $\lambda_{\mathrm{var}} = .01$, $\lambda_{\mathrm{smooth}} = 1$). We studied the effect of different temporal deformations on DTAN's ability to find the perturbed signals joint alignment and thus recover the latent input signals. Unlike in the UCR dataset (see below), in the synthetic dataset the latent source signal is available and can be used as a reference for evaluation. We studied the following aspects: (1) The difficulty of the input signals (Figure 4, the different columns); (2) the seriousness of the deformation, achieved by varying $K$, the dimension of the Dirichlet distribution

Table 2: Timing test-set alignments for a single-class synthetic data. There are 16 test sets. Within each set, the length of the signals is fixed. There are 4 different lengths (across the sets): 64, 128, 256, and 512. The size (*i.e.*, the number of signals) of each test set is either 10, $10^2$, $10^3$, or $10^4$. Taking all possible combinations of these 4 lengths and 4 sizes, yielded the 16 test sets. Each entry in the table represents the time it took to align an entire such test set by DTAN's forward pass.

| Alignment timing per test set (in [sec]) | | | | |
| --- | --- | --- | --- | --- |
| length \ # of signals | 10 | $10^2$ | $10^3$ | $10^4$ |
| 64 | 0.003 | 0.003 | 0.007 | 0.109 |
| 128 | 0.003 | 0.004 | 0.012 | 0.211 |
| 256 | 0.014 | 0.038 | 0.042 | 0.455 |
| 512 | 0.003 | 0.007 | 0.084 | 0.660 |

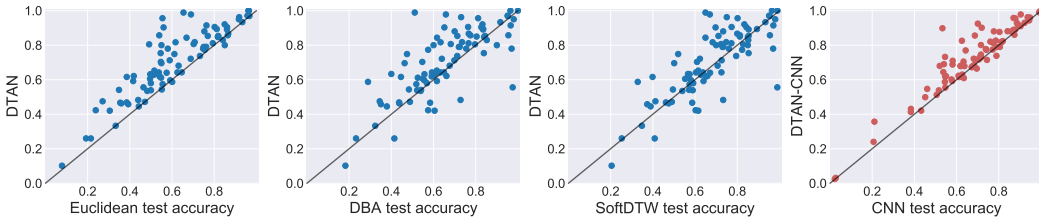

Figure 5: Correct classification rates using NCC. Each point above the diagonal indicates an entire UCR archive dataset [8] where DTAN achieved better (or no-worse) results than the competing method. Blue: DTAN's test accuracy compared with: Euclidean (DTAN was better or no worse in 93% of the datasets), DBA (77%) and SoftDTW (62%). Red: DTAN-CNN compared with CNN (87%).

(Table 1, rows) and (3) the number of recurrences (Figure 4, rows). We also measured the timings of alignment of a single-class test data by DTAN. The test sets vary in size ($10 : 10^4$, log-spaced values) and signal length (64, 128, 256, 512). We trained DTAN on 100 samples for each signal length. For each condition, we measured how long it took to align the entire test set via DTAN's forward pass. Timing was measured on a Nvidia GeForce GTX 1080 graphic card.

**Results.** Table 1 reports the average within-class variance of the misaligned signals ("Baseline") and the reduced variance after alignment by DTAN, R-DTAN2 and R-DTAN4 on both the train and test sets. The results show that DTAN generalizes well. In addition, as the number of diffeomorphic warps increases, R-DTAN performs finer alignments without increasing the number of parameters. Figure 4 illustrates how the synthetic misaligned signals are iteratively warped by R-DTAN, recovering the latent signals (up to a diffeomorphic offset). We also study the effect of adding Gaussian noise to the perturbed signals on DTAN's performance; see tables and discussion in the **Sup. Mat.** Table 2 summarizes the timing results, showing that DTAN's timing scales gracefully; *e.g.*, aligning the largest test set ($10^4$ signals of length 512) took DTAN only 0.66 [sec].

## 5.2 UCR Time-Series Classification Archive (Real Data)

The UCR time-series classification archive [8] contains 85 real-world datasets (we used 84). The datasets differ from each other in the number of examples, signal length, application domain (*e.g.*: ECG; medical imaging; motion sensors), and number of classes (2–60). We worked with the train and test sets provided with the archive. Here we report a summary of our results which appear in full detail (together with a study of the effect of the regularization term) at our **Sup. Mat.**

**Nearest Centroid Classification (NCC) experiment.** The 1-Nearest Neighbor (1-NN) classifier, when using the DTW distance, was shown [54, 5] to be on par with state-of-the-art time-series classifiers; however, 1-NN requires: 1) the entire train set to be stored; 2) DTW to be computed between each pair of training example and and test example. This scales poorly in terms of computational efficiency and storage. This issue is mitigated considerably by performing NCC, using each class average signal as a centroid [43]. In the lack of ground truth for the latent warps in real data, NCC success rates also provide an indicative metric for the quality of the joint alignment and/or average

signal. Thus, we perform NCC on the UCR archive, comparing DTAN to: (1) the sample mean of the misaligned sets (Euclidean); (2) DBA; (3) SoftDTW.

**Experiment outline.** For each of the UCR datasets, we trained DTAN in a similar fashion to 5.1, where $\lambda_{\mathrm{var}} \in [10^{-3}, 10^{-2}]$, $\lambda_{\mathrm{smooth}} \in [0.5, 1]$. We used R-DTANx, where $x \in \{1, 2, 4\}$ is the number of TT layers. We then computed the centroid (w.r.t. to a Euclidean distance) of each class in the aligned train set. NCC was conducted by aligning each test sample through the trained net and measuring a Euclidean distance to each of the centroids. DBA and SoftDTW were measured by DTW distance (which is the distance associated with these methods). We used Python's **tslearn**'s implementation of DTW, DBA and SoftDTW [51], limiting each to 100 iterations. The SoftDTW barycenter loss was minimized via L-BFGS [34] and the best $\gamma$ was chosen among the following values: $10^{-3}, 10^{-2}, 10^{-1}, 1$, and 10.

**Results.** Figure 5 shows the NCC experiment's results. Each point above the diagonal stands for an *entire dataset* where DTAN correct classification rate was better than (or equal to) the competing method. This was the case for 93% of the datasets when compared to Euclidean, 77% for DBA, and 62% for SoftDTW. These results (1) illustrate the importance of unwarping the misaligned data (as shown by the Euclidean case) and (2) indicate that averaging via DTAN under Euclidean geometry is usually superior to DTW-based averaging. These findings are also sup-

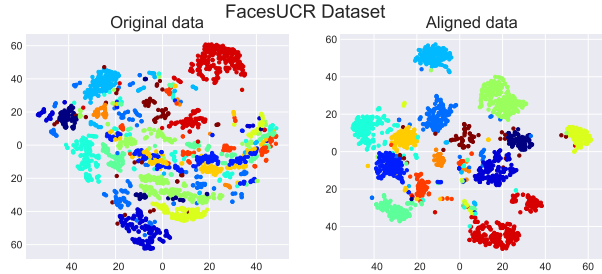

Figure 6: t-SNE visualization of the original and aligned test data of the 11-class FacesUCR dataset. The class labels are used here for visualization, but were not used during the test-data alignment. This highlights how DTAN decreases within-class variance while increasing inter-class variance.

ported by the average signals displayed in Figure 3. The Euclidean mean is strongly affected by the misalignment, while DBA falls to a bad local minimum. SoftDTW and DTAN show comparable qualitative results on this set, but note two major differences: (1) DTAN jointly aligns several classes within the same model (while SoftDTW had to be computed for each class separately) and (2) DTAN generalizes the learned alignment to new test samples (rightmost panel), while it is inapplicable for SoftDTW (as it must be computed again for new signals). For more results, please see our **Sup. Mat.**

**CNN classification experiment.** We also tested whether DTAN can increase CNN classification accuracy. We first trained DTAN to minimize Eq. (6) using the same regularization and recurrence parameters used in the NCC experiment. After training, we froze the weights of $f_{\mathrm{loc}}$ and fed DTAN's outputs to another CNN, and trained it for classification (identical to $f_{\mathrm{loc}}$ in terms of architecture and optimization). We call this model DTAN-CNN. Note other time-series averaging methods cannot be used in a similar way. We compared the average test accuracy of DTAN-CNN to the same CNN without DTAN, using 5 runs per dataset. DTAN-CNN achieved higher, or equal to, correct classification rates on 87% of the datasets (see Figure 5, red). Figure 6, which provides a **t-SNE** visualization of the original and aligned data [36], illustrates how DTAN decreases intra-class variance while increasing inter-class one, thus improving the performance of classification net.

# 6   Conclusion

Building on both recent ideas such as STN [28, 49], efficient highly-expressive diffeomorphisms [19, 20], and older ones such as congealing [31, 10], we proposed DTAN, a deep net for learning time-series joint alignment. The alignment learning is done in an unsupervised way. If, however, class labels are known in train time, we use them within a semi-supervised framework that reduces the variance within each class separately. In addition, we proposed a regularization term for the warps, which is critical in an unsupervised framework. We also proposed R-DTAN, a recurrent variant of DTAN, which improves the expressiveness and performance of DTAN without increasing the number of parameters. Our experiments showed that the proposed method works well on both training and test data sets.

**Acknowledgement:** NSD was supported by research grant #15334 from the VILLUM FONDEN.

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
