[Supplementary Material · supmat.pdf]

# Diffeomorphic Temporal Alignment Nets
—————
# Supplemental Material

**Ron Shapira Weber**
Ben-Gurion University
ronsha@post.bgu.ac.il

**Matan Eyal**
Ben-Gurion University
mataney@post.bgu.ac.il

**Nicki Skafte Detlefsen**
Technical University of Denmark
nsde@dtu.dk

**Oren Shriki**
Ben-Gurion University
shrikio@bgu.ac.il

**Oren Freifeld**
Ben-Gurion University
orenfr@cs.bgu.ac.il

## Contents

# 1 Temproal Transformer Nets

## 1.1 Model Architecture

As mentioned in our paper, a Temporal Transformer (TT) layer (Figure 1) is a variant of the Spatial Transformer layer [2], and is consisted of 3 modules:

**1. A localization network.** For an input signal, $U$, the localization network, $f_{\text{loc}}$, regresses the warp's parameters such that $f_{\text{loc}}(U) = \theta$.

**2. A parameterized grid generator**. This generator creates a discrete 1D grid of length $M$ (where $M$ is the length of $U$), $G = (p_m)_{m=1}^M \subset [-1, 1]$, of evenly-spaced points.

**3. A differentiable time-series resampler**. The output signal, $V$, is computed by interpolating the values of $V$ at $T^{\theta}(G)$ from $U$, as explained below.

Let $p_{i,m}^{\text{warped}} = T^{\theta_i}(p_m)$. The discrete-time $i$-th aligned signal is:

$$V_i = (V_{i,m})_{m=1}^M = (V_{i,1}, \ldots, V_{i,M}). \tag{1}$$

Note that due to the need to resample the signal, rather than having $V_i = U_i \circ T^{\theta_i}$, we must also account for the resampling kernel. For the popular linear kernel, we obtain (based on [2]),

$$V_{i,m} = \sum_{m'=1}^M U_{i,m'} \max(0, 1 - |p_{i,m}^{\text{warped}} - m'|). \tag{2}$$

To propagate the loss to the localization network, the resampling kernel must be differentiable, which is the case for the linear kernel used in this paper.

## 1.2 Derivatives

We provide the derivatives for the 1D Temporal Alignment Network (based on [2]). The derivative w.r.t. the parameterization of the warp family (*i.e.* the CPAB gradient) is discussed in the main paper.

$$\frac{\partial V_{i,m}}{\partial U_{i,m'}} = \max(0, 1 - |p_{i,m}^{\text{warped}} - m'|) \tag{3}$$

$$\frac{\partial V_{i,m}}{\partial (p_{i,m}^{\text{warped}})} = \sum_{m'=1}^M U_{i,m'} \begin{cases} 0 & \text{if} & |m' - p_{i,m}^{\text{warped}}| \geq 1 \\ 1 & \text{if} & m' \geq p_{i,m}^{\text{warped}} \\ -1 & \text{if} & m' < p_{i,m}^{\text{warped}} \end{cases} . \tag{4}$$

Where $V_{i,m}$ is the $i^{th}$ warped signal at time point $m$, $U_{i,m'}$ is the input signal at time point $m'$ and $p_{i,m}^{\text{warped}}$ is the $m^{th}$ point of the sampling grid. The generalization of these results to multichannel time series is straightforward and thus omitted.

Figure 1: The Diffeomorphic Temporal Transformer module. Figure adapted with permission from [4].

Table 1: Train and Test syntactic data experiments with an added Gaussian noise. The left tables corresponds to $\sigma = 0.01$ while the right one corresponds to $\sigma = 0.1$.

| $\sigma = 0.01$ | | | $\sigma = 0.1$ | | |
|---|---|---|---|---|---|
| Train set variance | | | Train set variance | | |
| Dataset | Baseline | DTAN4 | Dataset | Baseline | DTAN4 |
| $K = 32$ | 0.485 | **0.063** | $K = 32$ | 0.502 | **0.091** |
| $K = 16$ | 0.521 | **0.110** | $K = 16$ | 0.545 | **0.147** |
| $K = 8$ | 0.536 | **0.125** | $K = 8$ | 0.563 | **0.177** |
| Test set variance | | | Test set variance | | |
| $K = 32$ | 0.468 | **0.097** | $K = 32$ | 0.501 | **0.141** |
| $K = 16$ | 0.512 | **0.173** | $K = 16$ | 0.533 | **0.161** |
| $K = 8$ | 0.532 | **0.185** | $K = 8$ | 0.544 | **0.193** |

## 2 Synthetic Data Experimental Settings

The synthetic data generation procedure is as follows. First, we generated 4 input signals, each is a linear combination of 2 sinus signals of different frequencies. Next, we generated $n$ warped signals for each input signal in the following manner: we sampled a $K$-bin histogram from a symmetric $K$-dimensional Dirichlet distribution. For each such histogram, we calculated its Cumulative Distribution Function (CDF). Since a CDF is a monotonically non-decreasing function, it can be used as a (non-differentiable) warp from the unit interval into itself, while keeping the endpoints fixed. Thus, each CDF corresponds to random warp. We use these warps to generate the synthetic warp data. Note that as $K$ increases, each sample is more likely to be close to the uniformly distribution, which in turn means a close-to-linear CDF, *i.e.*, almost the identity warp. The value of $n$ was set to 250 unless stated otherwise. The network was trained for 2500 epochs with an Adam optimizer [3], learning rate $\eta = 10^{-4}$. The parameters of the smoothness prior were $\lambda_{val} = 0.01$ and $\lambda_{smooth} = 1$. The Partition, $\Omega$, was set such that $\dim(\boldsymbol{\theta}) = 32$. The Dirichlet distribution's $K$ values are either 8, 16, or 32, while the value of $\alpha$ (the parameter of the symmetric Dirichlet distribution) was fixed to $\alpha = 3$.

The train set contained 800 signals (200 per class) where the test set contained 200 signals. We used a 60-20-20% train, validation, and test split. After training, we computed the joint alignment of these sets by simply passing each through the network. As such, computing the average signal of each class is given by the arithmetic mean of each class of the aligned data.

**Synthetic Data with an Additive Gaussian Noise.** We expended our synthetic experiment (reported in the paper) by adding noise to increase the difficulty of the input data. We generated the warped signal similarly to the previous experiment, and then added an additional noise from a Gaussian distribution of mean $\mu = 0$ and a standard deviation of either $\sigma = 0.1$ or $\sigma = 0.01$ to the signal's values. Results are reported in Table 1.

## 3 Computing Infrastructures

We have used the following infrastructures in our experiments: Intel(R) Core(TM) i7-7740X CPU @ 4.30GHz, 8 cores, 16gb RAM with an Nvidia GeForce GTX 1080 graphic card.

# 4 Regularization effect on DTAN Joint Alignment

## 4.1 Regularization Effect

Figure 2: Regularization Effect on the ECGFiveDays(a)-(d) and Trace (e)-(h) datasets. Left: without the regularization term ($F_{reg}$). Right: with $F_{reg}$ ($\lambda_{var} = 0.1, \lambda_{smooth} = 1$). Since DTAN framework is unsupervised, $F_{data}$ could be minimized by unrealistically-large deformations; $F_{reg}$ prevents that.

## 4.2 With a Zero Boundary Condition on the CPA Velocity Field

Figure 3: Joint alignment of 10 randomly-chosen samples of a given class from the Trace dataset. $\lambda_{\text{var}}, \lambda_{\text{smooth}} \in [0.01, 0.1, 1, 10]$. $\dim(\boldsymbol{\theta}) = 8$.

## 4.3 Without a Zero Boundary Condition on the CPA Velocity Field

Figure 4: Removing the zero-boundary condition. Joint alignment of 10 randomly-chosen test samples of a given class from the Trace dataset. $\lambda_{\text{var}}, \lambda_{\text{smooth}} \in [0.01, 0.1, 1, 10]$. $\dim(\boldsymbol{\theta}) = 8$.

# 5 Additional Alignment Results of Test Data

## 5.1 Within-class Variance Reduction

Figure 5: Within-class joint alignment of several datasets from the UCR archive [1]. The presented results are of previously-unseen test samples. The red and blue signals indicate the misaligned and aligned data average signal, respectively. The shaded area stands for ± standard deviation.

## 5.2  Average Signal / Barycenters Comparison

(a) Plane Dataset

(b) yoga Dataset

(c) SwedishLeaf Dataset

## 5.3 Recurrent DTAN

Figure 7: ECG200 dataset. Top row: 10 Randomly-chosen test samples from each class of the dataset. Second row until last: RDTAN joint alignment recurrent process output at each stage. The blue line indicates the sample mean of the signals.

# 6 Nearest Centroid Classification (NCC)

We show here detailed results of the NCC experiment mentioned in the paper ("StarlightCurves" dataset was excluded). For the baseline experiment we used the Euclidean mean of the misaligned set. We compare between DTAN and time-series barycenter-averaging methods: DTW Barycenter Averaging (DBA) and SoftDTW.

Table 2: Nearest Centriod Classificaiton results.

| Dataset | Baseline | DTAN | Softdtw | DBA |
|---|---|---|---|---|
| 50words | 0.516484 | **0.652747** | 0.615385 | 0.615385 |
| Adiac | 0.549872 | **0.695652** | 0.501279 | 0.462916 |
| ArrowHead | 0.611429 | **0.748571** | 0.520000 | 0.474286 |
| Beef | 0.533333 | **0.633333** | 0.566667 | 0.400000 |
| BeetleFly | 0.850000 | 0.800000 | 0.850000 | **0.900000** |
| BirdChicken | 0.550000 | **0.800000** | 0.700000 | 0.600000 |
| CBF | 0.763333 | 0.914444 | **0.971111** | 0.965556 |
| Car | 0.616667 | **0.816667** | 0.683333 | 0.633333 |
| ChlorineConcentration | 0.333073 | 0.333073 | **0.348177** | 0.323698 |
| CinC ECG torso | 0.385507 | **0.615942** | 0.398551 | 0.445652 |
| Coffee | 0.964286 | **1.000000** | 0.964286 | 0.964286 |
| Computers | 0.416000 | 0.592000 | **0.640000** | 0.616000 |
| Cricket X | 0.238462 | 0.423077 | **0.602564** | 0.574359 |
| Cricket Y | 0.348718 | 0.541026 | **0.571795** | 0.541026 |
| Cricket Z | 0.305128 | 0.420513 | **0.615385** | 0.605128 |
| DiatomSizeReduction | 0.957516 | **0.970588** | 0.950980 | 0.950980 |
| DistalPhalanxOutlineAgeGroup | 0.817500 | 0.847500 | **0.850000** | 0.840000 |
| DistalPhalanxOutlineCorrect | 0.471667 | 0.471667 | **0.490000** | 0.488333 |
| DistalPhalanxTW | 0.747500 | **0.780000** | 0.760000 | 0.755000 |
| ECG200 | 0.750000 | **0.790000** | 0.730000 | 0.720000 |
| ECG5000 | 0.860444 | **0.891333** | 0.853778 | 0.834667 |
| ECGFiveDays | 0.689895 | **0.977933** | 0.670151 | 0.658537 |
| Earthquakes | 0.754658 | 0.773292 | **0.822981** | 0.574534 |
| ElectricDevices | 0.482687 | 0.534820 | **0.539748** | 0.538970 |
| FISH | 0.560000 | **0.902857** | 0.697143 | 0.651429 |
| FaceAll | 0.491716 | 0.804734 | **0.827811** | 0.796450 |
| FaceFour | 0.840909 | 0.829545 | **0.852273** | **0.852273** |
| FacesUCR | 0.539512 | **0.857073** | 0.812683 | 0.774634 |
| FordA | 0.495973 | **0.604832** | 0.552902 | 0.549570 |
| FordB | 0.499725 | 0.579758 | **0.591309** | 0.568482 |
| Gun Point | 0.753333 | **0.880000** | 0.733333 | 0.700000 |
| Ham | 0.761905 | **0.790476** | 0.733333 | 0.723810 |
| HandOutlines | 0.818000 | **0.850000** | 0.812000 | 0.804000 |
| Haptics | 0.392857 | **0.457792** | 0.373377 | 0.350649 |
| Herring | 0.546875 | **0.703125** | 0.609375 | 0.546875 |
| InlineSkate | 0.192727 | **0.260000** | 0.252727 | 0.232727 |
| InsectWingbeatSound | **0.601010** | 0.587374 | 0.328283 | 0.289394 |
| ItalyPowerDemand | 0.918367 | **0.962099** | 0.750243 | 0.730807 |
| LargeKitchenAppliances | 0.440000 | 0.482667 | **0.733333** | 0.728000 |
| Lighting2 | 0.688525 | **0.721311** | 0.622951 | 0.639344 |
| Lighting7 | 0.589041 | 0.712329 | **0.726027** | 0.698630 |
| MALLAT | 0.966738 | **0.968870** | 0.953945 | 0.952665 |
| Meat | **0.933333** | **0.933333** | **0.933333** | 0.916667 |
| MedicalImages | 0.385526 | **0.468421** | 0.461842 | 0.436842 |
| MiddlePhalanxOutlineAgeGroup | 0.732500 | 0.737500 | **0.795000** | 0.712500 |
| MiddlePhalanxOutlineCorrect | **0.551667** | 0.543333 | 0.495000 | 0.483333 |
| MiddlePhalanxTW | 0.591479 | **0.596491** | 0.581454 | 0.556391 |
| MoteStrain | 0.861022 | **0.904153** | 0.843450 | 0.826677 |
| NonInvasiveFatalECGThorax1 | 0.769466 | **0.853435** | 0.710941 | 0.712977 |
| NonInvasiveFatalECGThorax2 | 0.802036 | **0.905344** | 0.773028 | 0.763868 |
| OSULeaf | 0.359504 | 0.462810 | **0.475207** | 0.438017 |
| OliveOil | **0.866667** | **0.866667** | 0.800000 | 0.766667 |

Table 2: Nearest Centriod Classificaiton results.

| Dataset | Baseline | DTAN | Softdtw | DBA |
|---------|----------|------|---------|-----|
| PhalangesOutlinesCorrect | 0.625874 | **0.642191** | 0.637529 | 0.632867 |
| Phoneme | 0.078586 | 0.101793 | **0.204641** | 0.182489 |
| Plane | 0.961905 | **1.000000** | 0.990476 | **1.000000** |
| ProximalPhalanxOutlineAgeGroup | 0.819512 | **0.853659** | **0.853659** | 0.843902 |
| ProximalPhalanxOutlineCorrect | 0.646048 | 0.642612 | **0.725086** | 0.649485 |
| ProximalPhalanxTW | 0.707500 | **0.817500** | 0.747500 | 0.735000 |
| RefrigerationDevices | 0.354667 | 0.466667 | **0.586667** | 0.584000 |
| ScreenType | 0.442667 | **0.445333** | 0.389333 | 0.378667 |
| ShapeletSim | 0.500000 | 0.538889 | **0.588889** | 0.522222 |
| ShapesAll | 0.513333 | **0.628333** | **0.628333** | 0.603333 |
| SmallKitchenAppliances | 0.418667 | 0.621333 | 0.658667 | **0.661333** |
| SonyAIBORobotSurface | 0.811980 | **0.893511** | **0.893511** | 0.835275 |
| SonyAIBORobotSurfaceII | 0.793284 | **0.811123** | 0.772298 | 0.766002 |
| Strawberry | 0.668842 | **0.843393** | 0.649266 | 0.616639 |
| SwedishLeaf | 0.702400 | **0.806400** | 0.723200 | 0.681600 |
| Symbols | 0.864322 | 0.857286 | **0.954774** | **0.954774** |
| ToeSegmentation1 | 0.574561 | 0.640351 | **0.671053** | 0.614035 |
| ToeSegmentation2 | 0.546154 | 0.753846 | **0.853846** | 0.838462 |
| Trace | 0.580000 | 0.780000 | **0.970000** | **0.970000** |
| TwoLeadECG | 0.554873 | **0.956102** | 0.801580 | 0.811238 |
| Two Patterns | 0.464750 | 0.555750 | **0.989750** | 0.975000 |
| UWaveGestureLibraryAll | 0.849525 | **0.920715** | 0.833613 | 0.831937 |
| Wine | 0.555556 | **0.574074** | **0.574074** | 0.518519 |
| WordsSynonyms | 0.271160 | **0.474922** | 0.412226 | 0.344828 |
| Worms | 0.215470 | 0.259669 | 0.408840 | **0.414365** |
| WormsTwoClass | 0.541436 | 0.618785 | **0.651934** | 0.591160 |
| synthetic control | 0.916667 | 0.950000 | **0.980000** | **0.980000** |
| uWaveGestureLibrary X | 0.631212 | 0.681184 | **0.706868** | 0.676438 |
| uWaveGestureLibrary Y | 0.548297 | **0.611669** | 0.564768 | 0.525405 |
| uWaveGestureLibrary Z | 0.537409 | **0.642099** | 0.604132 | 0.592406 |
| wafer | 0.654445 | **0.988968** | 0.649416 | 0.511032 |
| yoga | 0.497000 | **0.631667** | 0.574000 | 0.557000 |

# 7 Alignment for CNN Classification

The alignment network (DTAN) was trained to minimize the within-class variance of each dataset (the training procedure is described in the main paper; "StarlightCurves" dataset was excluded). After the alignment phase is complete, DTAN output is fed into a subsequent classification network which is trained for time-series classification. During the classificaiton training phase, the weights of the alignment net are frozen. The two classification networks (with and without the DTAN) are identical in terms of architecture and number of parameters.

Table 3: Comparison between DTAN-CNN and baseline CNN on the UCR archive. The results are the mean classification test accuracy and standard deviation (SD) of 5 runs per dataset.

| Dataset | DTAN-CNN | SD | CNN | SD |
|---------|----------|-----|-----|-----|
| 50words | **0.7174** | 0.013778 | 0.7060 | 0.007483 |
| Adiac | **0.8266** | 0.005953 | 0.7858 | 0.017174 |
| ArrowHead | **0.6812** | 0.037172 | 0.6788 | 0.053604 |
| Beef | **0.2400** | 0.064464 | 0.2066 | 0.067896 |
| BeetleFly | **0.6200** | 0.107703 | 0.5500 | 0.044721 |
| BirdChicken | **0.6900** | 0.124097 | 0.5400 | 0.048990 |
| CBF | **0.8010** | 0.090988 | 0.7154 | 0.108776 |

Table 3: Comparison between DTAN-CNN and baseline CNN on the UCR archive. The results are the mean classification test accuracy and standard deviation (SD) of 5 runs per dataset.

| Dataset | DTAN-CNN | SD | CNN | SD |
|---|---|---|---|---|
| Car | **0.8800** | 0.019380 | 0.7766 | 0.029159 |
| ChlorineConcentration | 0.7384 | 0.016439 | **0.7890** | 0.010334 |
| CinCECGtorso | 0.6644 | 0.126010 | **0.6956** | 0.044992 |
| Coffee | **0.9286** | 0.039072 | 0.8858 | 0.035273 |
| Computers | **0.5384** | 0.032259 | 0.5160 | 0.028955 |
| CricketX | **0.6298** | 0.011583 | 0.6268 | 0.021255 |
| CricketY | **0.6732** | 0.015197 | 0.6722 | 0.020054 |
| CricketZ | **0.6230** | 0.013609 | 0.6094 | 0.038119 |
| DiatomSizeReduction | **0.9248** | 0.020556 | 0.7006 | 0.196982 |
| DistalPhalanxOutlineAgeGroup | **0.7454** | 0.073826 | 0.7154 | 0.056220 |
| DistalPhalanxOutlineCorrect | **0.8236** | 0.010744 | 0.8188 | 0.004214 |
| DistalPhalanxTW | 0.7890 | 0.005586 | **0.7904** | 0.006499 |
| ECG200 | **0.8900** | 0.017889 | 0.8780 | 0.019391 |
| ECG5000 | **0.8980** | 0.025020 | 0.5330 | 0.230003 |
| ECGFiveDays | **0.9012** | 0.085775 | 0.7190 | 0.115331 |
| Earthquakes | **0.8194** | 0.001200 | 0.8130 | 0.008626 |
| ElectricDevices | **0.6730** | 0.005621 | 0.6722 | 0.009108 |
| FISH | **0.9578** | 0.005810 | 0.8698 | 0.026806 |
| FaceAll | **0.7284** | 0.005463 | 0.7156 | 0.005713 |
| FaceFour | **0.7362** | 0.107300 | 0.7136 | 0.055770 |
| FacesUCR | **0.8888** | 0.019477 | 0.8696 | 0.017636 |
| FordA | 0.8602 | 0.008931 | **0.8742** | 0.002482 |
| FordB | **0.8168** | 0.010962 | 0.8130 | 0.033923 |
| GunPoint | **0.9548** | 0.035561 | 0.8788 | 0.053462 |
| Ham | **0.7162** | 0.047008 | 0.6914 | 0.024589 |
| HandOutlines | **0.8792** | 0.008280 | 0.8584 | 0.012816 |
| Haptics | **0.4986** | 0.006437 | 0.4514 | 0.003137 |
| Herring | **0.7096** | 0.018467 | 0.5814 | 0.018282 |
| InlineSkate | **0.3570** | 0.011610 | 0.2106 | 0.050294 |
| InsectWingbeatSound | 0.6092 | 0.006400 | **0.6344** | 0.013749 |
| ItalyPowerDemand | **0.9618** | 0.003311 | 0.9368 | 0.015992 |
| LargeKitchenAppliances | **0.5872** | 0.004750 | 0.5446 | 0.016354 |
| Lighting2 | **0.7050** | 0.023195 | 0.6330 | 0.039643 |
| Lighting7 | **0.6796** | 0.025255 | 0.6190 | 0.042100 |
| MALLAT | **0.8808** | 0.030485 | 0.8416 | 0.028793 |
| Meat | **0.8768** | 0.022649 | 0.8034 | 0.235758 |
| MedicalImages | **0.7278** | 0.011479 | 0.7104 | 0.010651 |
| MiddlePhalanxOutlineAgeGroup | 0.7146 | 0.064769 | **0.7440** | 0.017297 |
| MiddlePhalanxOutlineCorrect | **0.5544** | 0.008163 | 0.5466 | 0.007060 |
| MiddlePhalanxTW | **0.6154** | 0.011056 | 0.6122 | 0.005036 |
| MoteStrain | **0.8204** | 0.011253 | 0.7600 | 0.098574 |
| NonInvasiveFatalECGThorax1 | **0.9188** | 0.006242 | 0.8912 | 0.004534 |
| NonInvasiveFatalECGThorax2 | **0.9296** | 0.003262 | 0.9224 | 0.002577 |
| OSULeaf | **0.5924** | 0.017614 | 0.5794 | 0.024377 |
| OliveOil | **0.6800** | 0.162823 | 0.5200 | 0.146969 |
| PhalangesOutlinesCorrect | **0.7730** | 0.011883 | 0.7414 | 0.007116 |
| Phoneme | **0.0312** | 0.008232 | 0.0272 | 0.010778 |
| Plane | 0.9904 | 0.008499 | **0.9922** | 0.011285 |
| ProximalPhalanxOutlineAgeGroup | **0.7320** | 0.140671 | 0.5454 | 0.103219 |
| ProximalPhalanxOutlineCorrect | **0.9102** | 0.010833 | 0.9032 | 0.010068 |
| ProximalPhalanxTW | 0.7898 | 0.034061 | **0.8050** | 0.010159 |
| RefrigerationDevices | 0.4208 | 0.033707 | **0.4324** | 0.027543 |
| ScreenType | **0.4134** | 0.021878 | 0.3830 | 0.019453 |
| ShapeletSim | **0.5122** | 0.023464 | 0.5056 | 0.013185 |
| ShapesAll | **0.0248** | 0.009621 | 0.0230 | 0.017989 |
| SmallKitchenAppliances | **0.5468** | 0.052055 | 0.4608 | 0.034219 |
| SonyAIBORobotSurface | **0.6792** | 0.033325 | 0.6190 | 0.128223 |
| SonyAIBORobotSurfaceII | **0.7780** | 0.030926 | 0.7382 | 0.058253 |
| Strawberry | **0.9602** | 0.008134 | 0.9478 | 0.018627 |

Table 3: Comparison between DTAN-CNN and baseline CNN on the UCR archive. The results are the mean classification test accuracy and standard deviation (SD) of 5 runs per dataset.

| Dataset | DTAN-CNN | SD | CNN | SD |
|---|---|---|---|---|
| SwedishLeaf | **0.9488** | 0.005600 | 0.9424 | 0.012500 |
| Symbols | **0.6786** | 0.047416 | 0.6372 | 0.075470 |
| ToeSegmentation1 | **0.6878** | 0.037360 | 0.5808 | 0.066307 |
| ToeSegmentation2 | **0.7694** | 0.009646 | 0.6430 | 0.061410 |
| Trace | **0.9780** | 0.011662 | 0.9040 | 0.041280 |
| TwoLeadECG | **0.8558** | 0.099494 | 0.6866 | 0.080822 |
| TwoPatterns | **0.9964** | 0.001497 | 0.9954 | 0.001497 |
| UWaveGestureLibraryAll | 0.9434 | 0.005238 | **0.9440** | 0.003847 |
| Wine | **0.6444** | 0.099468 | 0.5372 | 0.051219 |
| WordsSynonyms | **0.6014** | 0.003826 | 0.5762 | 0.022746 |
| Worms | **0.4310** | 0.033728 | 0.3836 | 0.130716 |
| WormsTwoClass | **0.6188** | 0.016485 | 0.5458 | 0.054551 |
| syntheticcontrol | **0.8256** | 0.004454 | 0.6254 | 0.192905 |
| uWaveGestureLibraryX | **0.7748** | 0.005706 | 0.7712 | 0.005418 |
| uWaveGestureLibraryY | **0.6834** | 0.003720 | 0.6832 | 0.003868 |
| uWaveGestureLibraryZ | **0.7216** | 0.007552 | 0.6812 | 0.005344 |
| wafer | 0.9922 | 0.001327 | **0.9932** | 0.001470 |
| yoga | **0.8144** | 0.009871 | 0.7722 | 0.008612 |