[Reviews · NeurIPS 2019]

Reviewer 1



The work is original and a result of high-quality research. The method is very clearly explained and supported with nice and easy to understand visualizations. The proposed method is achieving impressive results on time series alignment and on time series classification when used as a feature extractor. Furthermore, the method can be used to align unseen data, at a fraction of the computational cost of other methods. Thus, the work is significant, with an impact on both the research field and industry.

Reviewer 2



EDIT: I've read the rebuttal and increase my score to 6. I still couldn't understand what you mean by "DBA/SDTW (but not DTAN) require test-data labels". It would be great to explain that better in the paper. Make sure you define all symbols before you use them and I'd recommend to give more background and context for your work in the introduction. ----- This paper proposes a deep learning method for time-series alignment and averaging. The key tool is a diffeomorphism T which allows to transform an input time-series U into a wrapped time-series V. On the plus side, the paper was enjoyable to read, includes several nice visualizations, and includes quite convincing experiments (although UCR contains only uni-dimensional time-series). On the negative side, I found that the notation was often not clearly defined, which made the proposed method hard to follow. For instance, AFAIK, \circ is never defined in the paper. The proposed method doesn't seem to handle variable-length time series, which I think is a very important feature. The technical methodology seems to follow mostly [43], although I don't think this is an issue if its use in a time-series context is novel. The proposed method includes several hyper-parameters to tune but DBA has zero and SDTW only one. I will adjust my score after the authors answered my questions below, in particular regarding notation, ability to handle variable-length time-series and computational cost of CPAB. General comments ---------------- * The introduction jumps right away into an equation. It would be better to provide some background and motivation for this work first. * The criticism that DTW and SDTW can't handle test data seems unfair, as it suffices to recompute an alignment between them. It would be fairer to say that your approach is a model-based, while DTW and SDTW are cast as an optimization problem. * It seems like the proposed method cannot compute alignments between variable-length time series but this was not entirely clear. Could you clarify? * Likewise, does the proposed approach handle multivariate time-series? The fact that DTW and SDTW are cast as an optimization problem allows to deal with variable-length time series. * Although it's probably not possible to describe it in full details, it would be great to explain how to compute CPAB and the gradient in more details. Could you also clarify the big O complexity? * Could you clarify how you choose the partition Omega in practice? Detailed comments ------------------ * Line 191: It was not clear why you claim that DTW and SDTW require class information. As I view it, this is not true. * Equations (1) and (2): what does \circ mean? It's not clear whether you mean element-wise multiplication between matrices or function composition. If they are matrices, please indicate the shape of U_i, V_i and W_i. * Instead of single vs. multiclass time-series, I would refer to unlabeled vs. labeled time-series. * Line 64: an optimal *monotonic* alignment * Line 66: The original DTW [41,42] is between a pair of times-series. Since you give an O(K^N) cost, I think you mean joint alignment of N time-series. Please provide a reference for this setting. * Equation (3): what is theta_i? Although it's meaning can be inferred, it should be explicitly defined. * Equation (3): Is the l.h.s. necessary? It is confusing, since it uses V_i while the r.h.s. uses (U_i, theta_i) (the signature of F is different on the l.h.s. and on the r.h.s.) * Equation (4): Typo: phi^theta -> T^theta * Line 157: I don't think you need to introduce the notation theta_i(U_i, w), theta_i = f_loc(U_i, w) is enough. Same for V_i. * Line 159: To be more explicit, maybe write "V_i = ..., where theta_i depends on w, as defined above".

Reviewer 3



I know little about time series, but the approach seems sound to me. In line with recent approaches for images and shape analysis. I think the authors should actually look at the recent approaches in 3D shape analysis to perform alignment via predicted deformations, such as Groueix, T., Fisher, M., Kim, V. G., Russell, B. C., & Aubry, M. (2018). 3d-coded: 3d correspondences by deep deformation. In Proceedings of the European Conference on Computer Vision (ECCV) (pp. 230-246). The philosophy of this type of approaches is very similar to the one of the paper, but the parametric deformation family T_\theta is learned by a neural network (an MLP), and thus could be better adapted to the signals. This would provide a nice comparison (i.e. CPAB performance v.s. MLP performance) I was a bit disappointed by the evaluations which is only on NN classification performance for the real data. I am not sure if there are better datasets that could be used. Anyhow, I think it would be good to analyse the results in much more details, for example correlating the performance/improvement of the proposed approach with the amount of data available for training or the variety of the data. === post-rebuttal I was quite disappointed by the rebuttal, which didn't address my comments for reasons I didn't agree with. I strongly encourage the authors to improve their experiments. However, I still think the paper is sound and quite original, so I do not oppose acceptance but don't strongly support it either.

[Author Response · NeurIPS 2019]

We thank the reviewers for their useful comments. As the most positive and confident reviewer, **R3**, raised no concerns (we were delighted to read **R3** has found our work significant and impactful), below we focus on **R4** and **R5**'s reviews.

**R4.** The reference for the complexity of joint alignment of N time-series via DTW is [4]. "∘" = function composition. Re Eq. (3): indeed, we define $\boldsymbol{\theta}_i$ only later (line 157); we'll fix this, thanks. As suggested, we will drop the LHS. **Eq. (4) is correct:** as written there, $T^{\boldsymbol{\theta}}(x) = \phi^{\boldsymbol{\theta}}(x, t = 1)$; note it is an **integral equation**; *i.e.*, its solution, $\phi^{\boldsymbol{\theta}}(x, \cdot)$ appears both outside and inside the integral (our notation is standard for such equations). Lines 157/159: agreed. Space limits prevent us from detailing how CPAB warp/gradient are evaluated; in short, the CPA structure lends itself to highly-efficient and highly-accurate solvers of the associated integral equations. For details, cf. [2, 1]. According to [2], the **CPAB complexity** is the sum of two **linear** terms, one w.r.t. # intervals in $\Omega$, and one w.r.t. the # of points in the signal, where the first term is negligible (unless the signal is very short); moreover, as points (and signals) are parallelized over via GPU, the proportionality constant of the 2nd term is small, yielding excellent timings (line 253). The complexity of each gradient component is similar and the components are parallelized over. True, DBA has no **Hyper-parameters (HPs)** and SDTW has 1 HP, but DBA clearly under-performs when compared with SDTW/DTAN, and, unlike DTAN, neither DBA nor SDTW generalizes. DTAN has 3 HPs: 1 for $\Omega$, and 2 for the prior. The effect of the last two is studied in our supmat. Other HPs (*e.g.*: the choice of $f_{\text{loc}}$; # of layers/neurons) are common in DL and are related to optimization and generalization capabilities of the model; **The choice of** $\Omega$ determines $\dim(\boldsymbol{\theta})$ (*i.e.*, the CPAB's expressiveness) and thus in practice we set it according to the training data availability: when data is scarce it is prudent to use a coarse $\Omega$ (hence low $\dim(\boldsymbol{\theta})$) to avoid over-fitting (note $\dim(\boldsymbol{\theta})$ = # of neurons in the last FC layer).

**R4. DBA/SDTW are applicable to test data only in the limited sense** that new optimization problems can be formulated and solved from scratch (see lines 3/32/70, and, especially, 196) and if we ignore the fact they need test-data labels; *i.e.*, we agree that for DBA/SDTW "it suffices to recompute an alignment" but note that (1) **on test data the difference in speed is huge** due to DTAN's fast forward pass (vs. DBA/SDTW's expensive computation of either DTW of each test signal to the train-set barycenter (BC) or a test-set BC) and (2) that, during test, DTAN does not need class labels; *i.e.*, for multi-class signals, DBA/SDTW/DTAN all require labels during training (as the alignment is within class). During test, however, only DBA/SDTW (but not DTAN) require test-data labels so they can recompute DTW between the new signal and the **correct** train-set BC. Without these labels, they must first solve an error-prone classification problem. We believe single-class/multi-class is the appropriate terminology (note that aligning together signals from different classes is usually undesirable). DTAN handles **Multivariate Data** easily, in the same manner Spatial Transformer Nets handles RGB images. **Variable-length (VL):** Our experiments focused on fixed-length signals. For both $f_{\text{loc}}$ and CPAB, VL is a non-issue: by dropping the boundary conditions (as done in the supmat), any time interval can be warped towards any other, even if they are of different lengths. We agree, however, that for VL the loss itself needs to be modified accordingly. In any case, please note SDTW had assumed a predefined fixed-length mean signal and they (like the authors of DBA) experimented only on datasets of within-dataset fixed lengths.

**R5.** As requested, we here add another evaluation using t-SNE visualization (Fig. 1). We do not assume data is available at large scale (though we **can and do** handle large data sets as well); *e.g.*, UCR contains 85 different datasets, many of which include only few exemplars per-class ('ECGFiveDays', Fig 1., main paper, had only $\sim 10$ samples per class). We believe our experiments section was extensive and thorough, and we refer the reviewer to our supmat which includes more analyses and results. The 1-NN experiment is the standard benchmark for time-series alignment/averaging (and is not used as a benchmark for best classification results). **We also included a CNN vs. DTAN-CNN evaluation** (lines 289-297); DBA/SDTW cannot be used for improving a CNN this way. We address works akin to [3] in lines 100-105; particularly, as [3] predicts **pairwise** warps between 3D shapes using templates, while DTAN learns joint-alignment of multiple 1D signals without one, the two methods are quite different.

Figure 1: t-SNE visualization of the original and aligned **test data** of the challenging 11-class 'FacesUCR' dataset. No class labels were used during DTAN alignment of the test data (it is used here only for visualization). The t-SNE highlights how DTAN decreases within-class variance while increasing inter-class variance. For DBA/SDTW, handling such multi-class test data alignment requires solving new optimization problems as well as (known or estimated) class labels of the test signals.

# References

[1] O. Freifeld. Deriving the CPAB derivative. Technical report, Ben-Gurion University, 2018.
[2] Freifeld et al. Transformations based on continuous piecewise-affine velocity fields. *IEEE TPAMI*, 2017.
[3] Groueix et al. 3d-coded: 3d correspondences by deep deformation. In *ECCV*, 2018.
[4] L. Wang and T. Jiang. On the complexity of multiple sequence alignment. *Journal of computational biology*, 1994.


[Meta-Review · NeurIPS 2019]

This paper developed a deep learning approach to aligning time series by incorporating a diffeomorphism. The reviewers found the paper enjoyable to read as the method was clearly explained and their were nice visualizations to present the intuition. The majority of the reviewers thought that the experiments were thorough enough to demonstrate the efficacy of the algorithm, however, one reviewer would have liked to see the method used for something beyond time-series classification. The author response did not really address this point to the reviewer's satisfaction, so the authors should consider this for the camera-ready version of the paper. Finally, the reviewers pointed out that the paper has some notational issues and the intro need some work to motivate the work and provide background. The authors should take these comments into account for the camera-read version to improve the quality of the paper.